# LEARNING CONTEXT-AWARE ADAPTIVE SOLVERS TO ACCELERATE CONVEX QUADRATIC PROGRAMMING

## ABSTRACT

Convex quadratic programming (QP) is an important sub-field of mathematical optimization. The alternating direction method of multipliers (ADMM) is a successful method to solve QP. Even though ADMM shows promising results in solving various types of QP, its convergence speed is known to be highly dependent on the step-size parameter $\rho$. Due to the absence of a general rule for setting $\rho$, it is often tuned manually or heuristically. In this paper, we propose CA-ADMM (Context-aware Adaptive ADMM)) which learns to adaptively adjust $\rho$ to accelerate ADMM. CA-ADMM extracts the spatio-temporal context, which captures the dependency of the primal and dual variables of QP and their temporal evolution during the ADMM iterations. CA-ADMM chooses $\rho$ based on the extracted context. Through extensive numerical experiments, we validated that CA-ADMM effectively generalizes to unseen QP problems with different sizes and classes (i.e., having different QP parameter structures). Furthermore, we verified that CA-ADMM could dynamically adjust $\rho$ considering the stage of the optimization process to accelerate the convergence speed further.

## 1 INTRODUCTION

Among the optimization classes, quadratic program (QP) is widely used due to its mathematical tractability, e.g. convexity, in various fields such as portfolio optimization (Boyd et al., 2017; Cornuéjols et al., 2018; Boyd et al., 2014; Markowitz, 1952), machine learning (Kecman et al., 2001; Sha et al., 2002), control, (Buijs et al., 2002; Krupa et al., 2022; Bartlett et al., 2002), and communication applications (Luo & Yu, 2006; Hons, 2001). As the necessity to solve large optimization problems increases, it is becoming increasingly important to ensure the scalability of QP for achieving the solution of the large-sized problem accurately and quickly.

Among solution methods to QP, first-order methods (Frank & Wolfe, 1956) owe their popularity due to their superiority in efficiency over other solution methods, for example active set (Wolfe, 1959) and interior points methods (Nesterov & Nemirovskii, 1994). The alternating direction method of multipliers (ADMM) (Gabay & Mercier, 1976; Mathematique et al., 2004) is commonly used for returning high solution quality with relatively small computational expense (Stellato et al., 2020b). Even though ADMM shows satisfactory results in various applications, its convergence speed is highly dependent on both parameters of QP and user-given step-size $\rho$. In an attempt to resolve these issues, numerous studies have proposed heuristic (Boyd et al., 2011; He et al., 2000; Stellato et al., 2020a) or theory-driven (Ghadimi et al., 2014)) methods for deciding $\rho$. But a strategy for selecting the best performing $\rho$ still needs to be found (Stellato et al., 2020b). Usually $\rho$ is tuned in a case-dependent manner (Boyd et al., 2011; Stellato et al., 2020a; Ghadimi et al., 2014).

Instead of relying on hand-tuning $\rho$, a recent study (Ichnowski et al., 2021) utilizes reinforcement learning (RL) to learn a policy that adaptively adjusts $\rho$ to accelerate the convergence of ADMM. They model the iterative procedure of ADMM as the Markov decision process (MDP) and apply the generic RL method to train the policy that maps the current ADMM solution states to a scalar value of $\rho$. This approach shows relative effectiveness over the heuristic method (e.g., OSQP (Stellato et al., 2020a)), but it has several limitations. It uses a scalar value of $\rho$ that cannot adjust $\rho$ differently depending on individual constraints. And, it only considers the current state without its history to determine $\rho$, and therefor cannot capture the non-stationary aspects of ADMM iterations. This method inspired us to model the iteration of ADMM as a non-stationary networked system. We

developed a more flexible and effective policy for adjusting $\rho$ for all constraints simultaneously and considering the evolution of ADMM states.

In this study, we propose Context-aware Adaptive ADMM (CA-ADMM), an RL-based adaptive ADMM algorithm, to increase the convergence speed of ADMM. To overcome the mentioned limitations of other approaches, we model the iterative solution-finding process of ADMM as the MDP whose context is determined by QP structure (or parameters). We then utilize a graph recurrent neural network (GRNN) to extract (1) the relationship among the primal and dual variables of the QP problem, i.e., its spatial context and (2) the temporal evolutions of the primal and dual variables, i.e., its temporal context. The policy network then utilizes the extracted spatio-temporal context to adjust $\rho$.

From the extensive numerical experiments, we verified that CA-ADMM adaptively adjusts $\rho$ in consideration of QP structures and the iterative stage of the ADMM to accelerate the convergence speed further. We evaluated CA-ADMM in various QP benchmark datasets and found it to be significantly more efficient than the heuristic and learning-based baselines in number of iterations until convergence. CA-ADMM shows remarkable generalization to the change of problem sizes and, more importantly, benchmark datasets. Through the ablation studies, we also confirmed that both spatial and temporal context extraction schemes are crucial to learning a generalizable $\rho$ policy. The contributions of the proposed method are summarized below:

- **Spatial relationships:** We propose a heterogeneous graph representation of QP and ADMM state that captures spatial context and verifies its effect on the generalization of the learned policy.

- **Temporal relationships:** We propose to use a temporal context extraction scheme that captures the relationship of ADMM states over the iteration and verifies its effect on the generalization of the learned policy.

- **Performance/Generalization:** CA-ADMM outperforms state-of-the-art heuristics (i.e., OSQP) and learning-based baselines on the training QP problems and, more importantly, out-of-training QP problems, which include large size problems from a variety of domains.

## 2 RELATED WORKS

**Methods for selecting $\rho$ of ADMM.** In the ADMM algorithm, step-size $\rho$ plays a vital role in determining the convergence speed and accuracy. For some special cases of QP, there is a method to compute optimal $\rho$ (Ghadimi et al., 2014). However, this method requires linear independence of the constraints, e.g., nullity of $\boldsymbol{A}$ is nonzero, which is difficult to apply in general QP problems. Thus, various heuristics have been proposed to choose $\rho$ (Boyd et al., 2011; He et al., 2000; Stellato et al., 2020a). Typically, the adaptive methods that utilize state-dependent step-size $\rho_t$ show a relatively faster convergence speed than non-adaptive methods. He et al. (2000); Boyd et al. (2011) suggest a rule for adjusting $\rho_t$ depending on the ratio of residuals. OSQP (Stellato et al., 2020a) extends the heuristic rule by adjusting $\rho$ with the values of the primal and dual optimization variables. Even though OSQP shows improved performance, designing such adaptive rules requires tremendous effort. Furthermore, the designed rule for a specific QP problem class is hard to generalize to different QP classes having different sizes, objectives and constraints. Recently, Ichnowski et al. (2021) employed RL to learn a policy for adaptively adjusting $\rho$ depending on the states of ADMM iterations. This method outperforms other baselines, showing the potential that an effective rule for adjusting $\rho$ can be learned without problem-specific knowledge using data. However, this method still does not sufficiently reflect the structural characteristics of the QP and the temporal evolution of ADMM iterations. Both limitations make capturing the proper problem context challenging, limiting its generalization capability to unseen problems of different sizes and with alternate objectives and constraints.

**Graph neural network for optimization problems.** An optimization problem comprises objective function, decision variables, and constraints. When the optimization variable is a vector, there is typically interaction among components in the decision vector with respect to an objective or constraints. Thus, to capture such interactions, many studies have proposed to use graph representation to model such interactions in optimization problems. Gasse et al. (2019) expresses mixed integer programming (MIP) using a bipartite graph consisting of two node types, decision variable

nodes, and constraint nodes. They express the relationships among variables and the relationships between decision variables and constraints using different kinds of edges whose edge values are associated with the optimization coefficients. Ding et al. (2019) extends the bipartite graph with a new node type, 'objective'. The objective node represents the linear objective term and is connected to the variable node, with the edge features and coefficient value of the variable component in the objective term. They used the tripartite graph as an input of GCN to predict the solution of MIP.

**ML accelerated scientific computing.** ML models are employed to predict the results of complex computations, e.g., the solution of ODE and PDE, and the solution of optimization problems, and trained by supervised learning. As these methods tend to be sample inefficient, other approaches learn operators that expedite computation speeds for ODE (Poli et al., 2020; Berto et al., 2021), fixed point iterations (Bai et al., 2021; Park et al., 2021b), and matrix decomposition (Donon et al., 2020). Poli et al. (2020) uses ML to predict the residuals between the accurate and inaccurate ODE solvers and use the predicted residual to expedite ODE solving. Bai et al. (2021); Park et al. (2021b) trains a network to learn alternate fixed point iterators, trained to give the same solution while requiring less iterations to converge. However, the training objective all the listed methods is minimizing the network prediction with the ground truth solution, meaning that those methods are at risk to find invalid solutions. CA-ADMM learns to provide the proper $\rho$ that minimize ADMM iterations, rather than directly predicting the solution of QP. This approach, i.e., choose $\rho$ for ADMM, still guarantees that the found solution remains equal to the original one.

## 3 BACKGROUND

**Quadratic programming (QP)** A quadratic program (QP) associated with $N$ variables and $M$ constraints is an optimization problem having the form of the following:

$$\min_x \quad \frac{1}{2} x^\top \boldsymbol{P} x + \boldsymbol{q}^\top x \tag{1}$$
$$\text{subject to} \quad \boldsymbol{l} \leq \boldsymbol{A} x \leq \boldsymbol{u},$$

where $x \in \mathbb{R}^N$ is the decision variable, $\boldsymbol{P} \in \mathbb{S}_+^N$ is the positive semi-definite cost matrix, $\boldsymbol{q} \in \mathbb{R}^N$ is the linear cost term, $\boldsymbol{A} \in \mathbb{R}^{M \times N}$ is a $M \times N$ matrix that describes the constraints, and $\boldsymbol{l}$ and $\boldsymbol{u}$ are their lower and upper bounds. QP generally has no closed-form solution except some special cases, for example $\boldsymbol{A} = \boldsymbol{0}$. Thus, QP is generally solved via some iterative methods.

**First-order QP solver** Out of the iterative methods to solve QP, the alternating direction method of multipliers (ADMM) has attracted considerable interest due to its simplicity and suitability for various large-scale problems including statistics, machine learning, and control applications. (Boyd et al., 2011). ADMM solves a given QP($\boldsymbol{P}, \boldsymbol{q}, \boldsymbol{l}, \boldsymbol{A}, \boldsymbol{u}$) through the following iterative scheme. At each step $t$, ADMM solves the following system of equations:

$$\begin{bmatrix} P + \sigma I & A^T \\ A & \text{diag}(\rho)^{-1} \end{bmatrix} \begin{bmatrix} x_{t+1} \\ \nu_{t+1} \end{bmatrix} = \begin{bmatrix} \sigma x_t - q \\ z_t - \text{diag}(\rho)^{-1} y_t \end{bmatrix}, \tag{2}$$

where $\sigma > 0$ is the regularization parameter that assures the unique solution of the linear system, $\rho \in \mathbb{R}^M$ is the step-size parameter, and $\text{diag}(\rho) \in \mathbb{R}^{M \times M}$ is a diagonal matrix with elements $\rho$. By solving Eq. (2), ADMM finds $x_{t+1}$ and $\nu_{t+1}$. Then, it updates $y_t$ and $z_t$ with the following equations:

$$\tilde{z}_{t+1} \leftarrow z_t + \text{diag}(\rho)^{-1}(\nu_{t+1} - y_t) \tag{3}$$
$$z_{t+1} \leftarrow \Pi(\tilde{z}_t + \text{diag}(\rho)^{-1} y_t) \tag{4}$$
$$y_{t+1} \leftarrow x_t + \text{diag}(\rho)(\tilde{z}_{t+1} - z_{t+1}), \tag{5}$$

where $\Pi$ is the projection onto the hyper box $[\boldsymbol{l}, \boldsymbol{u}]$. ADMM proceeds the iteration until the primal $r_t^{\text{primal}} = \boldsymbol{A} x_t - z_t \in \mathbb{R}^M$ and dual residuals $r_t^{\text{dual}} = \boldsymbol{P} x_t + \boldsymbol{q} + \boldsymbol{A}^\top y_t \in \mathbb{R}^N$ are sufficiently small. For instance, the termination criteria are as follows:

$$||r_t^{\text{primal}}||_\infty \leq \epsilon^{\text{primal}}, ||r_t^{\text{dual}}||_\infty \leq \epsilon^{\text{dual}}, \tag{6}$$

where $\epsilon^{\text{primal}}$ and $\epsilon^{\text{dual}}$ are sufficiently small positive numbers.

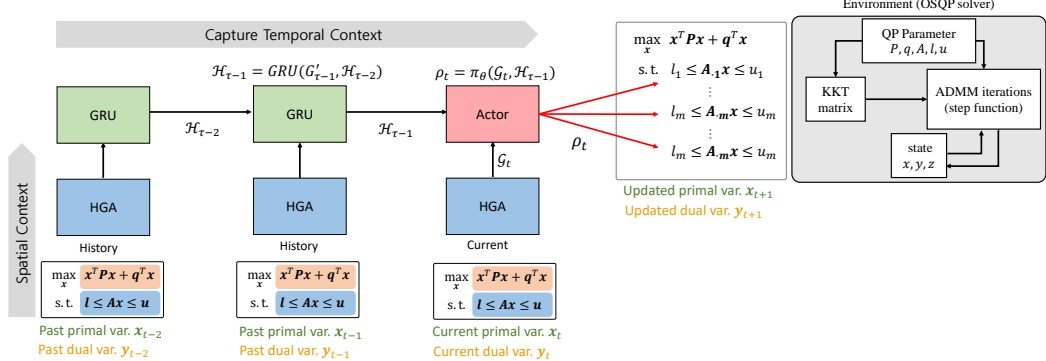

Figure 1: **An overview of CA-ADMM**. CA-ADMM uses the spatio-temporal context, which is extracted by GRNN from the ADMM state observations, to compute $\rho$ for ADMM algorithm.

## 4 METHOD

In this section, we present $\underline{C}$ontext-aware $\underline{A}$daptive ADMM (CA-ADMM) that automatically adjusts $\rho$ parameters considering the spatial and temporal contexts of the ADMM to accelerate the convergence of ADMM. We first introduce the contextual MDP formulation of ADMM iterations for solving QP problems and discuss two mechanisms devised to extract the spatial and temporal context from the ADMM iterations.

### 4.1 CONTEXTUAL MDP FORMULATION

As shown in Eqs. (2) to (5), ADMM solves QP by iteratively updating the intermediate variables $x_t$, $y_t$, and $z_t$. Also, the updating rule is parameterized with QP variables $\boldsymbol{P}, \boldsymbol{q}, \boldsymbol{l}, \boldsymbol{A}, \boldsymbol{u}$.

We aim to learn a QP solver that can solve general QP instances whose structures are different from QPs used for training the solver. As shown in Eqs. (2) to (5), ADMM solves QP by iteratively updating the intermediate variables $x_t$, $y_t$, and $z_t$. Also, the updating rule is parameterized with QP variables $\boldsymbol{P}, \boldsymbol{q}, \boldsymbol{l}, \boldsymbol{A}, \boldsymbol{u}$. In this perspective, we can consider the iteration of ADMM for each QP as a different MDP whose dynamics can be characterized by the structure of QP and the ADMM's updating rule. Based on this perspective, our objective, learning a QP solver that can solve general QP instances, is equivalent to learning an adaptive policy (ADMM operator) that can solve a contextual MDP (QP instance with different parameters).

Assuming we have a QP($\boldsymbol{P}, \boldsymbol{q}, \boldsymbol{A}, \boldsymbol{l}, \boldsymbol{u}$), where $\boldsymbol{P}, \boldsymbol{q}, \boldsymbol{A}, \boldsymbol{l}, \boldsymbol{u}$ are the parameters of the QP problem, and ADMM($\phi$), where $\phi$ is the parameters of the ADMM algorithm. In our study, $\phi$ is $\rho$, but it can be any configurable parameter that affects the solution updating procedure. We then define the contextual MDP $\mathcal{M} = \big(\mathbb{X}, \mathbb{U}, T_\rho^c, R, \gamma\big)$, where $\mathbb{X} = \{(x_t, y_t, z_t)\}_{t=1,2,\ldots}$ is the set of states, $\mathbb{U} = \{\rho_t\}_{t=1,2,\ldots}$ is the set of actions, $T_\rho^c(x, x') = P(x_{t+1} = x'|x_t = x, \rho_t = \rho, c)$ is the transition function whose behaviour is contextualized with the context $c = \{\boldsymbol{P}, \boldsymbol{q}, \boldsymbol{A}, \boldsymbol{l}, \boldsymbol{u}, \phi\}$, $R$ is the reward function that returns 0 when $x_t$ is the terminal state (i.e., the QP problem is solved) and -1 otherwise, and $\gamma \in [0, 1)$ is a discount factor. We set an MDP transition made at every 10 ADMM steps to balance computational cost and acceleration performance. That is, we observe MDP states at every 10 ADMM steps and change $\rho_t$; thus, during the ten steps, the same actions $\rho_t$ are used for ADMM. As shown in the definition of $T_\rho^c$, the context $c$ alters the solution updating procedure of ADMM (i.e., $T_\rho^c \neq T_\rho^{c'}$ to the given state $x$). Therefore, it is crucial to effectively process the context information $c$ in deriving the policy $\pi_\theta(x_t)$ to accelerate the ADMM iteration.

One possible approach to accommodate $c$ is manually designing a feature that serves as a (minimal) sufficient statistic that separates $T_\rho^c$ from the other $T_\rho^{c'}$. However, designing such a feature can involve enormous efforts; thus, it derogates the applicability of learned solvers to the new solver classes. Alternatively, we propose to learn a context encoder that extracts the more useful context information from not only the structure of a target QP but also the solution updating history of the ADMM so that we can use context-dependent adaptive policy for adjusting $\rho$. To successfully

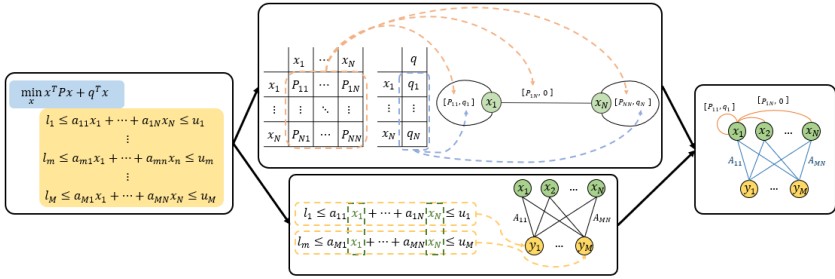

Figure 2: **Heterograph representation of QP**.

extract the context information from the states, we consider the relationships among the primal and dual variables (i.e., the spatial relationships) and the evolution of primal/dual variables with ADMM iteration (i.e., temporal relationship). To consider such spatio-temporal correlations, we propose an encoder network that is parameterized via a GNN followed by RNN. Fig. 1 visualizes the overall network architecture.

## 4.2 Extracting Spatial Context Information via Heterogeneous Graph Neural Network

**Heterogeneous graph representation of QP.** The primal $x_t$ and dual $y_t$ variables of QP are related via the QP parameters $(\boldsymbol{P}, \boldsymbol{q}, \boldsymbol{A}, \boldsymbol{l}, \boldsymbol{u})$. As shown in Fig. 2, their relationship can be modeled as a heterogeneous graph where the nodes represent the primal and dual variables, and the edges represent the relationship among these variables. As the roles of primal and dual variables in solving QP are distinctive, it necessitates the graph representation that reflects the different variable type information. Regarding that, we represent the relationships of the variables at $t$-th MDP step with a directed heterogeneous graph $\mathcal{G}_t = (\mathbb{V}_t, \mathbb{E}_t)$. As the graph construction is only associated with the current $t$, we omit the step-index $t$ for notational brevity. The node set $\mathbb{V}$ consist of the primal $\mathbb{V}_{\text{primal}}$ and dual node sets $\mathbb{V}_{\text{dual}}$. The $n$-th primal and $m$-th dual nodes have node feature $s_n^{\text{primal}}$ and $s_m^{\text{dual}}$ defined as:

$$s_n^{\text{primal}} = [\underbrace{\log_{10} r_n^{\text{dual}}, \log_{10} ||r^{\text{dual}}||_\infty, \mathbb{1}_{r_n^{\text{dual}} < \epsilon^{\text{dual}}}}_{\text{Encoding ADMM state}}], \tag{7}$$

$$s_m^{\text{dual}} = [\underbrace{\log_{10} r_m^{\text{primal}}, \log_{10} ||r^{\text{primal}}||_\infty, \mathbb{1}_{r_m^{\text{primal}} < \epsilon^{\text{primal}}}, y_m, \rho_m, \min(z_m - l_m, u_m - z_m)}_{\text{Encoding ADMM state}}$$
$$\underbrace{\mathbb{1}_{\text{equality}}, \mathbb{1}_{\text{inequality}}}_{\text{Encoding QP problem}}], \tag{8}$$

where $r_m^{\text{primal}}$ and $r_n^{\text{dual}}$ denotes the $m$-th and $n$-th element of primal and dual residuals, respectively, $|| \cdot ||_\infty$ denotes the infinity norm, $\mathbb{1}_{r_m^{\text{primal}} < \epsilon^{\text{primal}}}$ and $\mathbb{1}_{r_n^{\text{dual}} < \epsilon^{\text{dual}}}$ are the indicators whether the $m$-th primal and $n$-th dual residual is smaller than $\epsilon^{\text{primal}}$ and $\epsilon^{\text{dual}}$, respectively. $\mathbb{1}_{\text{equality}}$ and $\mathbb{1}_{\text{inequality}}$ are the indicators whether the $m$-th constraint is equality and inequality, respectively.

The primal and dual node features are design to capture ADMM state information and the QP structure. However, those node features do not contain information of $\boldsymbol{P}$, $\boldsymbol{q}$, or $\boldsymbol{A}$. We encode such QP problem structure in the edges. The edge set $\mathbb{E}$ consist of

- The primal-to-primal edge set $\mathbb{E}^{\text{p2p}}$ is defined as $\{e_{ij}^{\text{p2p}} | \boldsymbol{P}_{ij} \neq 0 \,\forall (i,j) \in [\![1, N]\!] \times [\![1, N]\!]\}$ where $[\![1, N]\!] = \{1, 2, \dots, N\}$. i.e., the edge from the $i$ th primal node to $j$ th primal node exists when the corresponding $\boldsymbol{P}$ is not zero. The edge $e_{ij}^{\text{p2p}}$ has the feature $s_{ij}^{\text{p2p}}$ as $[\boldsymbol{P}_{ij}, \boldsymbol{q}_i]$.

- The primal-to-dual edge set $\mathbb{E}^{\text{p2d}}$ is defined as $\{e_{ij} | \boldsymbol{A}_{ji} \neq 0 \,\forall (i,j) \in [\![1, N]\!] \times [\![1, M]\!]\}$. The edge $e_{ij}^{\text{p2d}}$ has the feature $s_{ij}^{\text{p2d}}$ as $[\boldsymbol{A}_{ji}]$.

- The dual-to-primal edge set $\mathbb{E}^{\text{d2p}}$ is defined as $\{e_{ij}|\boldsymbol{A}_{ij} \neq 0 \,\forall (i,j) \in [\![1,M]\!] \times [\![1,N]\!]\}$. The edge $e_{ij}^{\text{d2p}}$ has the feature $s_{ij}^{\text{d2p}}$ as $[\boldsymbol{A}_{ij}]$.

**Graph preprocessing for unique QP representation.** QP problems have the same optimal decision variables for scale transformation of the objective function and constraints. However, such transformations alter the QP graph representation. To make graph representation invariant to the scale transform, we preprocess QP problems by dividing the elements of $\boldsymbol{P}, \boldsymbol{q}, \boldsymbol{a}, \boldsymbol{A}, \boldsymbol{u}$ with problem-dependent constants. The details of the preprocessing steps are given in Appendix A.

**Grpah embedding with Hetero GNN.** The heterogeneous graph $\mathcal{G}$ models the various type of information among the primal and dual variables. To take into account such type information for the node and edge embedding, we propose heterogeneous graph attention (HGA), a variant of type-aware graph attention (Park et al., 2021a), to embed $\mathcal{G}$. HGA employs separate node and edge encoders for each node and edge type. For each edge type, it computes edge embeddings, applies the attention mechanism to compute the weight factors, and then aggregates the resulting edge embeddings via weighted summation. Then, HGA sums the different-typed aggregated edge embeddings to form a type-independent edge embedding. Lastly, HGA updates node embeddings by using the type-independent edge embeddings as an input to node encoders. We provide the details of HGA in Appendix B.

### 4.3 Extracting temporal context via RNN

The previous section discusses extracting context by considering the spatial correlation among the variables within the MDP step. We then discuss extracting the temporal correlations. As verified from numerous optimization literature, deciding $\rho$ based on the multi-stage information (e.g., momentum methods) helps increase the stability and solution quality of solvers. Inspired by that, we consider the time evolution of the ADMM states to extract temporal context. To accomplish that we use an RNN to extract the temporal context from the ADMM state observations. At the $t^{\text{th}}$ MDP step, we consider $l$ historical state observations to extract the temporal context. Specifically, we first apply HGA to $\mathcal{G}_{t-l}, \ldots, \mathcal{G}_{t-1}$ and then apply GRU (Chung et al., 2014) as follows:

$$\mathcal{G}'_{t-l+\delta} = \text{HGA}(\mathcal{G}_{t-l+\delta}) \qquad \text{for } \delta = 0, 1, 2, \cdots, l-1 \qquad (9)$$

$$\mathcal{H}_{t-l+\delta} = \text{GRU}(\mathbb{V}'_{t-l+\delta}, \mathcal{H}_{t-l+\delta-1}) \qquad \text{for } \delta = 1, 2, \cdots, l-1 \qquad (10)$$

where $\mathcal{G}'_{t-l+\delta}$ is updated graph at $t-l+\delta$, $\mathbb{V}'_{t-l+\delta}$ is the set of updated nodes of $\mathcal{G}'_{t-l+\delta}$, and $\mathcal{H}_{t-l+\delta}$ are the hidden embeddings. We define the node-wise context $\mathcal{C}_t$ at time $t$ as the last hidden embedding $\mathcal{H}_{t-1}$.

### 4.4 Adjusting $\rho_t$ using extracted context

We parameterize the policy $\pi_\theta(\mathcal{G}_t, \mathcal{C}_t)$ via the HGA followed by an MLP. Since action $\rho_t$ is not defined at $t$, we exclude $\rho_t$ from the node features of $\mathcal{G}_t$ and instead node-wise concatenate $\mathcal{C}_t$ to the node features. For the action-value network $Q_\phi(\mathcal{G}_t, \rho_t, \mathcal{C}_t)$, we use $\mathcal{G}_t$, $\rho_t$, and $\mathcal{C}_t$ as inputs. $Q_\phi(\mathcal{G}_t, \rho_t, \mathcal{C}_t)$ has a similar network architecture as $\pi_\theta(\mathcal{G}_t, \mathcal{C}_t)$. We train $\pi_\theta(\mathcal{G}_t, \mathcal{C}_t)$ and $Q_\phi(\mathcal{G}_t, \rho_t, \mathcal{C}_t)$ on the QP problems of size 10~15 (i.e., $N \in [10, 15]$) with DDPG (Lillicrap et al., 2015) algorithm. Appendix C details the used network architecture and training procedure further.

**Comparison to `RLQP`.** Our closest baseline `RLQP` (Ichnowski et al., 2021) is also trained using RL. At step $t$, it utilizes the state $x = (\{s_m^{\text{dual}}\}_{m=1,\ldots,M})$, with

$$s_m^{\text{dual}} = [\log_{10}||r^{\text{primal}}||_\infty, \log_{10}||r^{\text{dual}}||_\infty], y_m, \rho_m,$$
$$\min(z_m - l_m, u_m - z_m), z_m - (Ax)_m]. \qquad (11)$$

It then uses an shared MLP $\pi_\theta(s_n^{\text{dual}})$ to compute $\rho_n$. We observe that the state representation and policy network are insufficient to capture the contextual information for differentiating MDPs. The state representation does not include information of $\boldsymbol{P}$ and $\boldsymbol{q}$, and the policy network does not explicitly consider the relations among the dual variables. However, as shown in Eq. (2), ADMM iteration results different $(x_t, \nu_t)$ with changes in $\boldsymbol{P}$, $\boldsymbol{q}$ and $\boldsymbol{A}$, meaning that $T_\rho^c$ also changes. Therefore, such state representations and network architectures may result in an inability to capture contextual

Table 1: **In-domain results (in # iterations)**. Smaller is better ($\downarrow$). Best in **bold**. We measure the average and standard deviation of ADMM iterations for each QP benchmark. All QP problems are generated to have $10 \leq N \leq 15$.

|  | Random QP | EqQP | Porfolio | SVM | Huber | Control | Lasso |
|---|---|---|---|---|---|---|---|
| OSQP | $81.12 \pm 21.93$ | $25.52 \pm 0.73$ | $185.18 \pm 28.73$ | $183.79 \pm 40.88$ | $54.38 \pm 5.37$ | $30.38 \pm 24.98$ | $106.17 \pm 45.70$ |
| RLQP | $69.42 \pm 27.35$ | $20.27 \pm 0.49$ | $28.45 \pm 5.52$ | $34.14 \pm 3.84$ | $25.16 \pm 0.66$ | $26.78 \pm 15.84$ | $25.61 \pm 0.68$ |
| CA-ADMM | $\mathbf{27.44 \pm 4.86}$ | $\mathbf{19.61 \pm 0.61}$ | $\mathbf{20.12 \pm 0.47}$ | $\mathbf{25.05 \pm 2.30}$ | $\mathbf{20.32 \pm 0.69}$ | $\mathbf{17.52 \pm 3.88}$ | $\mathbf{19.08 \pm 1.01}$ |

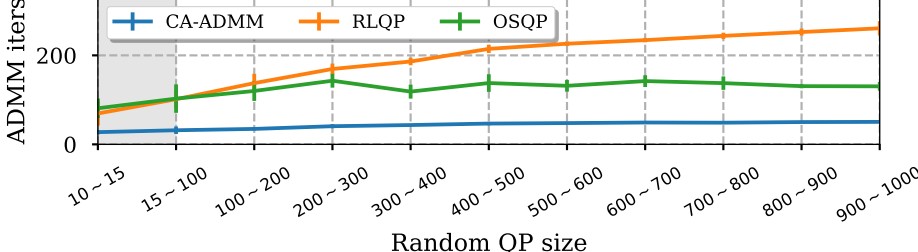

Figure 3: **QP size generalization results**. Smaller is better ($\downarrow$). We plot the average and standard deviation of the iterations as the bold lines and error bars, respectively. The gray area (■) indicates the training QP sizes.

changes among QP problems. Additionally, using single state observation as the input for $\pi_\theta$ may also hinder the possibility of capturing temporal context.

## 5 EXPERIMENTS

We evaluate CA-ADMM in various canonical QP problems such as `Random QP`, `EqQP`, `Porfolio`, `SVM`, `Huber`, `Control`, and `Lasso`. We use `RLQP`, another learned ADMM solver (Ichnowski et al., 2021), and `OSQP`, a heuristic-based solver (Stellato et al., 2020a), as baselines. Training and testing problems are generated according to Appendix D.

**In-domain results.** It is often the case that a similar optimization is repeatedly solved with only parameter changes. We train the solver using a particular class of QP instances and apply the trained solver to solve the same class of QP problems but with different parameters. We use the number of ADMM iterations until it's convergence as a performance metric for quantifying the solver speed. Table 1 compares the averages and standard deviations of the ADMM iterations required by different baseline models in solving the QP benchmark problem sets. As shown in Table 1, CA-ADMM exhibits $2 \sim 8$x acceleration compared to `OSQP`, the one of the best heuristic method. We also observed that `CA-ADMM` consistently outperforms `RLQP` with a generally smaller standard deviations.

We evaluate the size-transferability and scalability of CA-ADMM by examining whether the models trained using smaller QP problems still perform well on the large QPs. We trained CA-ADMM using small-sized Random QP instances having $10 \sim 15$ variables and applied the trained solve to solve large-scaled Random QP instances having up to $1000$ variables. Fig. 3 shows how the number of ADMM iterations varies with the size of QP. As shown in the figure, `CA-ADMM` requires the least iterations for all sizes of test random QP instances despite having been trained on the smallest setting of $10 \sim 15$. The gap between our method and `RLQP` becomes larger as the QP size increases. It is interesting to note that `OSQP` requires relatively few ADMM iterations when the problem size becomes larger. It is possibly because the parameters of `OSQP` are tuned to solve the medium-sized QPs. We conclude that CA-ADMM learn how to effectively adjust ADMM parameters such that the method can effectively solve even large-sized QP instances.

**Cross-domain results.** We hypothesize that our learned QP solvers can transfer to new domains more easily than other baselines. To test this hypothesis, we train `CA-ADMM` and `RLQP` on `Random QP` and `EqQP` datasets, which do not assume specific QP parameter values or constraints. The trained models are evaluated on different QP domains without additional training, i.e., which is equivalent to a zero-shot transfer setting. As shown in Table 2, `CA-ADMM` consistently outperforms `RLQP` in absolute ADMM steps and shows less performance degradation when transferring to new domains.

Table 2: **Cross-domain results. zero-shot transfer (`Random QP + EqQP`) → X, where X ∈** {`Random QP`, `EqQP`, `Porfolio`, `SVM`, `Huber`, `Control`, `Lasso`}: Smaller is better (↓). Best in **bold**. We measure the average and standard deviation of ADMM iterations for each QP benchmark. The gray colored cell (▨) denotes the training problems. Green colored values are the case where the transferred model outperforms the in-domain model. All QP problems are generated to have $40 \leq N + M \leq 70$.

| | Random QP | EqQP | Porfolio | SVM | Huber | Control | Lasso |
|---|---|---|---|---|---|---|---|
| OSQP | $160.99 \pm 92.21$ | $25.52 \pm 0.73$ | $185.18 \pm 28.73$ | $183.79 \pm 40.88$ | $54.38 \pm 5.37$ | $30.38 \pm 24.98$ | $106.17 \pm 45.70$ |
| RLQP (transferred) | $65.28 \pm 57.75$ | $28.48 \pm 0.70$ | $31.80 \pm 7.97$ | $37.96 \pm 10.16$ | $28.70 \pm 0.67$ | $74.60 \pm 49.71$ | $29.18 \pm 29.18$ |
| RLQP (in-domain) | $69.42 \pm 27.35$ | $20.27 \pm 0.49$ | $28.45 \pm 5.52$ | $34.14 \pm 3.84$ | $25.16 \pm 0.66$ | $26.78 \pm 15.84$ | $25.61 \pm 0.68$ |
| CA-ADMM (transferred) | $\mathbf{28.52} \pm 7.13$ | $\mathbf{19.38} \pm 0.64$ | $\mathbf{25.38} \pm 7.03$ | $\mathbf{31.26} \pm 13.44$ | $\mathbf{19.56} \pm 0.92$ | $\mathbf{27.16} \pm 4.97$ | $\mathbf{23.84} \pm 2.10$ |
| CA-ADMM (in-domain) | $27.44 \pm 4.86$ | $19.61 \pm 0.61$ | $20.12 \pm 0.47$ | $25.05 \pm 2.30$ | $20.32 \pm 0.69$ | $17.52 \pm 3.88$ | $19.08 \pm 1.01$ |

We also observed that for `EqQP` and `Huber` `CA-ADMM` exhibits better performance than in-domain cases. The experiment results indicate that the context encoder plays a crucial role in transferring to different problem classes.

**Benchmark QP results.** From the above results, we confirm that CA-ADMM outperforms the baseline algorithms in various synthetic datasets. We then evaluate CA-ADMM on the 134 public QP benchmark instances (Maros & Mészáros, 1999), which have different distributions of generating QP parameters (i.e., $P, q, A, l$ and $u$). For these tasks, we trained CA-ADMM and `RLQP` on `Random QP`, `EqQP`, `Portfolio`, `SVM`, `Huber`, `Control` and `Lasso`. As shown in Table A.8, CA-ADMM exhibits the lowest iterations for 85 instances. These results indicate that CA-ADMM can generalize to the QP problems that follow completely different generating distributions.

**Application to linear programming.** Linear programming (LP) is an important class of mathematical optimization. By setting $P$ as the zero matrix, a QP problem becomes an LP problem. To further understand the generalization capability of CA-ADMM, we apply CA-ADMM trained with $10 \leq N < 15$ `Random QP` instances to solve 100 random LP of size $10 \leq N < 15$. As shown in Table A.7, we observed that CA-ADMM shows $\sim 1.75$ and $\sim 30.78$ times smaller iterations than `RLQP` and `OSQP` while solving all LP instances within 5000 iterations. These results again highlight the generalization capability of CA-ADMM to the different problem classes.

**Computational times.** We measure the performance of solver with the number of iteration to converge (i.e., solve) as it is independent from the implementation and computational resources. However, having lower computation time is also vital. With a desktop computer equips an AMD Threadripper 2990WX CPU and Nvidia Titan X GPU, we measure the compuational times of CA-ADMM, `OSQP` on GPU and `OSQP` on CPU. At each MDP step, CA-ADMM and the baseline algorithms compute $\rho$ via the following three steps: (1) Configuring the linear system and solving it – *linear system solving*, (2) constructing the state (e.g., graph construction) – *state construction*, and (3) computing $\rho$ from the state – *$\rho$ computation*. We first measure the average unit computation times (e.g., per MDP step computational time) of the three steps on the various-sized QP problems. As shown in Table A.5, the dominating factor is *linear system solving* step rather than *state construction* and *$\rho$ computation* steps. We then measure the total computation times of each algorithm. As shown Table A.6, when the problem size is small, CA-ADMM takes longer than other baselines, especially OSQP. However, as the problem size increases, CA-ADMM takes less time than other baselines due to the reduced number of iterations.

# 6 ABLATION STUDY

**Effect of spatial context extraction methods.** To understand the contribution of spatial and temporal context extraction schemes to the performance of our networks, we conduct an ablation study. The variants are `HGA` that uses spatial context extraction schemes and `RLQP` that does not use a spatial context extracting scheme. As shown in Table 3, the models using the spatial context (i.e., `CA-ADMM`

Table 3: **Ablation study results.** Best in **bold**. We measure the average and standard deviation of ADMM iterations.

| | Components | | | Problem size ($N$) | | | | | | | | | | |
| --- | --- | --- | --- | --- | --- | --- | --- | --- | --- | --- | --- | --- | --- | --- |
| | Graph repr. | HGA | GRU | $10 \sim 15$ | $15 \sim 100$ | $100 \sim 200$ | $200 \sim 300$ | $300 \sim 400$ | $400 \sim 500$ | $500 \sim 600$ | $600 \sim 700$ | $700 \sim 800$ | $800 \sim 900$ | $900 \sim 1000$ |
| RLQP | ✗ | ✗ | ✗ | 69.42 ± 27.35 | 101.44 ± 20.00 | 137.60 ± 20.97 | 169.40 ± 12.16 | 186.20 ± 8.42 | 214.60 ± 9.24 | 226.20 ± 5.19 | 234.20 ± 4.83 | 243.80 ± 7.08 | 252.40 ± 7.86 | 260.80 ± 16.15 |
| HGA | ✓ | ✓ | ✗ | **25.96** ± 5.11 | 31.96 ± 7.17 | 36.90 ± 2.66 | 43.10 ± 3.81 | 45.40 ± 3.07 | 49.80 ± 2.82 | 50.40 ± 1.96 | 51.40 ± 2.76 | 52.80 ± 1.89 | 53.30 ± 2.79 | 55.20 ± 2.44 |
| CA-ADMM | ✓ | ✓ | ✓ | 27.44 ± 4.86 | **31.70** ± 8.29 | **34.70** ± 3.07 | **40.80** ± 4.60 | **43.40** ± 2.24 | **46.70** ± 3.69 | **47.80** ± 2.56 | **49.10** ± 1.97 | **48.70** ± 2.00 | **50.10** ± 1.44 | **50.50** ± 1.28 |

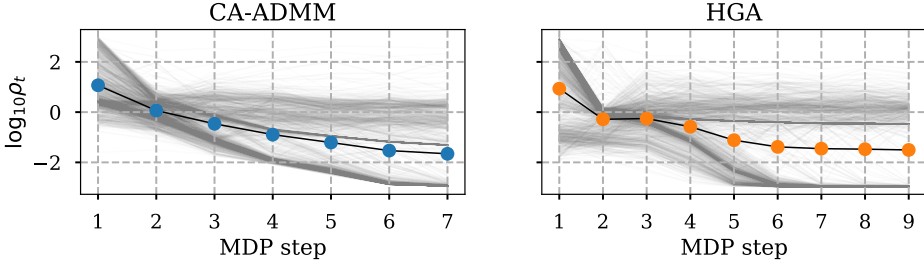

Figure 4: **Example of** $\log_{10} \rho_t$ **over MDP steps on a QP of** $N = 2,000$ **and** $M = 6000$. The gray lines visualize the time evolution of $\log_{10} \rho_t$. The black lines and markers visualize the averages of $\log_{10} \rho_t$ over the dual variables. CA-ADMM terminates at the 6-th MDP (i.e., solving QP in 60 ADMM steps). HGA takes additional two MDP steps to solve the given QP.

and HGA) outperform the model without the spatial context extraction. The results indicate that the spatial context significantly contributes to higher performance.

**Effect of temporal context extraction methods.** To further investigate the effects of temporal context, we evaluate CA-ADMM, HGA that exclude GRU (i.e., temporal context extraction scheme) from our model. We also consider RLQP. Table 3 shows that for $N \leq 15$ HGA produces better performance than CA-ADMM, while for $N \geq 15$ the opposite is true. We conclude that the temporal-extraction mechanism is crucial to generalizing to larger QP problems.

**Qualitative analysis.** Additionally, we visualize the $\rho_t$ of CA-ADMM and HGA during solving a QP problem size of 2,000. As shown in Fig. 4, both of the models gradually decrease $\rho_t$ over the course of optimization. This behavior is similar to the well-known step-size scheduling heuristics implemented in OSQP. We also observed that $\rho_t$ suggested by CA-ADMM generally decreases over the optimization steps. On the other hand, HGA shows the plateau on scheduling $\rho_t$. We observed similar patterns from the different QP instances with different sizes.

**Effect of MDP step interval.** We update $\rho$ every $N = 10$ ADMM iteration. In principle, we can adjust $\rho$ at every iteration while solving QP. In general, a smaller $N$ allows the algorithm to change $\rho$ more often and potentially further decrease the number of iterations. However, it is prone to increase the computation time due to frequent linear system solving (i.e., solving Eq. (1)). Having higher $N$ shows the opposite tendency in the iteration and computational time trade-off. Table Table A.9 summarizes the effect of $N$ on the ADMM iterations and computational times. From the results, we confirm that $N = 10$ attains a nice balance of iterations and computational time.

## 7 CONCLUSION

To enhance the convergence property of ADMM, we introduced CA-ADMM, a learning-based policy that adaptively adjusts the step-size parameter $\rho$. We employ a spatio-temporal context encoder that extracts the spatial context, i.e., the correlation among the primal and dual variables, and temporal context, i.e., the temporal evolution of the variables, from the ADMM states. The extracted context contains information about the given QP and optimization procedures, which are used as input to the policy for adjusting $\rho$ for the given QP. We empirically demonstrate that CA-ADMM significantly outperforms the heuristics and learning-based baselines and that CA-ADMM generalizes to the different QP class benchmarks and large-size QPs more effectively.

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

## A   GRAPH PREPROCESSING

In this section, we provide the details of the graph preprocessing scheme that imposes the scale invariant to the heterograph representation. The preprocessing step is done by scaling the objective function, and the constraints. Algorithm 1 and 2 explain the preprocessing procedures.

---

**Algorithm 1:** Objective scaling

**Input:** Quadratic cost matrix $\boldsymbol{P} \in \mathbb{S}_+^N$, linear cost terms $q \in \mathbb{R}^N$
**Output:** Scaled quadratic cost matrix $\boldsymbol{P}' \in \mathbb{S}_+^N$, scale linear cost terms $q' \in \mathbb{R}^N$

1  **for** $n = 1, 2, ...$ **do**
2      $p^* = \max(\max_n(|\boldsymbol{P}_{nn}|), 2\max_{(i,j), i \neq j} |\boldsymbol{P}_{ij}|)$          `// Compute scaler`
3      $\boldsymbol{P}' = \boldsymbol{P}/p^*$
4      $\boldsymbol{q}' = \boldsymbol{q}/p^*$
5  **end**

---

**Algorithm 2:** Constraints scaling

**Input:** constraint matrix $\boldsymbol{A} \in \mathbb{R}^{M \times N}$, lower bounds $\boldsymbol{l} \in \mathbb{R}^M$, upper bound $\boldsymbol{u} \in \mathbb{R}^M$.
**Output:** Scaled constraint matrix $\boldsymbol{A}' \in \mathbb{R}^{M \times N}$, Scaled lower bounds $\boldsymbol{l}' \in \mathbb{R}^M$,
Scaled upper bound $\boldsymbol{u}' \in \mathbb{R}^M$.

6  **for** $m = 1, 2, ...$ **do**
7      $a_m^* = \max_n(|\boldsymbol{A}_{mn}|)$          `// Compute scaler for each row`
8      $\boldsymbol{A}'_m = \boldsymbol{A}_m/a_m^*$
9      $\boldsymbol{l}'_m = \boldsymbol{l}_m/a_m^*$
10      $\boldsymbol{u}'_m = \boldsymbol{u}_m/a_m^*$
11  **end**

---

## B   DETAILS OF HGA

In this section, we provide the details of the HGA layer.

HGA is composed of edge update, edge aggregation, node update and hetero node aggregation to extract the informative context. This is done by considering each edge type and computing the updated node, edge embedding.

We denote source and destination nodes with the index $i$ and $j$ and the edge type with $k$. $h_i$ and $h_j$ denote the source and destination node embedding and $h_{ij}^k$ denotes the embedding of the $k$ type edge between the node $i$ and $j$.

**Edge update** HGA computes the edge embedding $h'_{ij}$ and the corresponding attention logit $z_{ij}$ as:

$$h'_{ij} = \text{HGA}_E^k([h_i, h_j, h_{ij}^k]) \tag{A.1}$$

$$z_{ij} = \text{HGA}_A^k([h_i, h_j, h_{ij}^k]), \tag{A.2}$$

where $\text{HGA}_E^k$ and $\text{HGA}_A^k$ are the hetero edge update function and the hetero attention function for edge type $k$, respectively. Both functions are parameterized with Multilayer Perceptron (MLP).

**Edge aggregation** HGA aggregates the typed edges as follows:

$$\alpha_{ij} = \frac{\exp(z_{ij}^k)}{\sum_{i \in N_k(j)} \exp(z_{ij}^k)} \tag{A.3}$$

where $N_k(j) = \{v_i | \text{type of } e_{ij} = k, v_j \in \mathcal{N}(i) \; \forall i\}$ is the set of all neighborhood of $v_j$, which type of edge from the element of the set to $v_j$ is $k$.
Second, HGA aggregates the messages and produces the type $k$ message $m_j^k$ for node $v_j$.

$$m_j^k = \sum_{i \in N_k(j)} \alpha_{ij} h'_{ij} \tag{A.4}$$

Table A.1: **The MLP sturcture of functions in HGA's first layer**

| | input dimensions | hidden dimensions | output dimensions | hidden activations | output activation |
|---|---|---|---|---|---|
| $\mathrm{HGA}_E^{\mathrm{p2p}}$ | 8 | | | | |
| $\mathrm{HGA}_E^{\mathrm{p2d}}$ | 12 | | | | |
| $\mathrm{HGA}_E^{\mathrm{d2p}}$ | 12 | | | | |
| $\mathrm{HGA}_A^{\mathrm{p2p}}$ | 8 | | | | |
| $\mathrm{HGA}_A^{\mathrm{p2d}}$ | 12 | $[64, 16]$ | 8 | LeakyReLU | |
| $\mathrm{HGA}_A^{\mathrm{d2p}}$ | 12 | | | | |
| $\mathrm{HGA}_V^{\mathrm{p2p}}$ | 11 | | | | |
| $\mathrm{HGA}_V^{\mathrm{p2d}}$ | 16 | | | | |
| $\mathrm{HGA}_V^{\mathrm{d2p}}$ | 11 | | | | |
| $\mathrm{HGA}_N$ | 8 | $[64, 32]$ | | | |

**Edge type-aware node update** The aggregated message with edge type $k$, $m_j^k$, is used to compute the updated node embedding with edge type $k$, $\mathfrak{h}_j^{k'}$, as:

$$\mathfrak{h}_j^k = \mathrm{HGA}_V^k([h_j, m_j^k]) \tag{A.5}$$

where $\mathrm{HGA}_V^k$ is the hetero node update function, which is composed of MLP.

The above three steps (edge update, edge aggregation, edge type-aware node update) are performed separately on each edge type to preserve the characteristics of the edge type and extract meaningful embeddings.

**Hetero Node update** HGA aggregates the updated hetero node feature $\mathfrak{h}_j^k$ to produce the updated node embedding $h_j'$ as follows.
First, HGA computes the attention logit $d_j^k$ corresponding to $\mathfrak{h}_j^k$ as:

$$d_j^k = \mathrm{HGA}_N(\mathfrak{h}_j^k) \tag{A.6}$$

where $\mathrm{HGA}_N$ is the hetero node update function, which is composed of MLP.
Second, HGA computes attention $\beta_j^k$ using the softmax function with $d_j^k$:

$$\beta_j^k = \frac{\exp(d_j^k)}{\sum_{k \in E_{dst}(j)} \exp(d_j^k)} \tag{A.7}$$

where $E_{dst}(j)$ is the set of edge types that has a destination node at $j$.
Finally, HGA aggregates the per-type the updated messages to compute updated node feature $h_j'$ for $v_j$ as:

$$h_j' = \sum_{k \in E_{dst}(j)} \beta_j^k \mathfrak{h}_j^k \tag{A.8}$$

## C  DETAILS OF NETWORK ARCHITECTURE AND TRAINING

In this section, we provide the architecture of pocliy network $\pi_\theta$ and action-value network $Q_\phi$.

**Policy netowrk $\pi_\theta$ architecture.**  As explained in Section 4.4, the policy network is consist HGA and MLP. We parameterize $\pi_\theta$ as follows:

$$\rho_t = \mathrm{MLP}_\theta\Big(\mathrm{HGA}_\theta\big(\mathrm{CONCAT}(\mathcal{G}_t, \mathcal{C}_t)\big)\Big), \tag{A.9}$$

Table A.2: **The MLP sturcture of functions in HGA's $n$ th layer($n \geq 2$)**

| | input dimensions | hidden dimensions | output dimensions | hidden activations | output activation |
|---|---|---|---|---|---|
| $\text{HGA}_E^{\text{p2p}}$ | | | | | |
| $\text{HGA}_E^{\text{p2d}}$ | | | | | |
| $\text{HGA}_E^{\text{d2p}}$ | | | | | |
| $\text{HGA}_A^{\text{p2p}}$ | | | | | |
| $\text{HGA}_A^{\text{p2d}}$ | 8 | $[64, 16]$ | 8 | LeakyReLU | |
| $\text{HGA}_A^{\text{d2p}}$ | | | | | |
| $\text{HGA}_V^{\text{p2p}}$ | | | | | |
| $\text{HGA}_V^{\text{p2d}}$ | | | | | |
| $\text{HGA}_V^{\text{d2p}}$ | | | | | |
| $\text{HGA}_N$ | | $[64, 32]$ | | | |

where $\text{MLP}_\theta$ is two-layered MLP with the hidden dimension $[64, 32]$, the LeakyReLU hidden activation, the ExpTanh (Eq. (A.10)) last activation, and $\text{HGA}_\theta$ is a HGA layer.

$$\text{ExpTanh}(x) = (\tanh(x) + 1) \times 3 + (-3) \tag{A.10}$$

**Q network $Q_\pi$ architecture.** As mentioned in Section 4.4, $Q_\pi$ has similar architecture to $\pi_\theta$. We parameterize $Q_\pi$ as follows:

$$\rho_t = \text{MLP}_\phi\Big(\text{READOUT}\Big(\text{HGA}_\phi\big(\text{CONCAT}(\mathcal{G}_t, \mathcal{C}_t, \rho_t)\big)\Big)\Big), \tag{A.11}$$

where $\text{MLP}_\phi$ is two-layered MLP with the hidden dimension $[64, 32]$, the LeakyReLU hidden activation, the Identity last activation, $\text{HGA}_\phi$ is a HGA layer, and READOUT is the weighted sum, min, and amx readout function that summarizes node embeddings into a single vector.

**Training detail.** We train all models with mini-batches of size 128 for 5,000 epochs by using Adam with the fixed learning rate. For $\pi_\theta$, we set learning rate as $10^{-4}$ and, for $Q_\phi$, we set learning rate as $10^{-3}$. We set history length $l$ as 3.

## D    DETAILS OF PROBLEM GENERATION

In this section, we provide the generation scheme of problem classes in OSQP, which is used to train and test our model. Every problem class is based on OSQP (Stellato et al., 2020a), but we modified some settings, such as the percentage of nonzero elements in the matrix or matrix size, to match graph sizes among the problem classes.

### D.1    RANDOM QP

$$\begin{aligned}\min_{x} \quad & \frac{1}{2}x^\top \boldsymbol{P}x + \boldsymbol{q}^\top x \\ \text{subject to} \quad & \boldsymbol{l} \leq \boldsymbol{A}x \leq \boldsymbol{u},\end{aligned} \tag{A.12}$$

**Problem instances** We set a random positive semidefinite matrix $\boldsymbol{P} \in \mathbb{R}^{n \times n}$ by using matrix multiplication on random matrix $\boldsymbol{M}$ and its transpose, whose element $\boldsymbol{M}_{ij} \sim \mathcal{N}(0, 1)$ has only 15% being nonzero elements. We generate the constraint matrix $x \in \mathbb{R}^{m \times n}$ with $\boldsymbol{A}_{ij} \sim \mathcal{N}(0, 1)$ with only 45% nonzero elements. We chose upper bound $\boldsymbol{u}$ with $\boldsymbol{u}_i \sim (Ax)_i + \mathcal{N}(0, 1)$, where $x$ is randomly chosen vector, and the upper bound $\boldsymbol{l}_i$ is same as $-\infty$.

## D.2 EQUALITY CONSTRAINED QP

$$\min_x \quad \frac{1}{2}x^\top \boldsymbol{P}x + \boldsymbol{q}^\top x$$
$$\text{subject to} \quad \boldsymbol{l} = \boldsymbol{A}x = \boldsymbol{u}, \tag{A.13}$$

**Problem instances** We set a random positive semidefinite matrix $\boldsymbol{P} \in \mathbb{R}^{n \times n}$ by using matrix multiplication on random matrix $\boldsymbol{M}$ and its transpose, whose element $\boldsymbol{M}_{ij} \sim \mathcal{N}(0,1)$ has only 30% of being nonzero elements. We generate the constraint matrix $\boldsymbol{A} \in \mathbb{R}^{m \times n}$ with $\boldsymbol{A}_{ij} \sim \mathcal{N}(0,1)$ with only 75% nonzero elements. We chose upper bound $\boldsymbol{u}$ with $\boldsymbol{u}_i \sim (Ax)_i$, where $x$ is randomly chosen vector, and the upper bound $\boldsymbol{l}_i \in \boldsymbol{l}$ is same as $\boldsymbol{u}_i$.

## D.3 PORTFOLIO OPTIMIZATION

$$\min_x \quad \frac{1}{2}x^\top Dx + \frac{1}{2}\boldsymbol{y}^\top \boldsymbol{y} - \frac{1}{2\gamma}\mu^\top x$$
$$\text{subject to} \quad \boldsymbol{y} = \boldsymbol{F}^\top x, \tag{A.14}$$
$$\mathbf{1}^\top x = 1,$$
$$x \geq 0,$$

**Problem instances** We set the factor loading matrix $\boldsymbol{F} \in \mathbb{R}^{n \times m}$, whose element $\boldsymbol{F}_{ij} \sim \mathcal{N}(0,1)$ as 90% nonzero elements. We generate a diagonal matrix $\boldsymbol{D} \in \mathbb{R}^{n \times n}$, whose element $\boldsymbol{D}_{ii} \sim \mathcal{C}(0,1) \times \sqrt{m}$ is the asset-specific risk. We chose a random vector $\mu \in \mathbb{R}^n$, whose element $\mu_i \sim \mathcal{N}(0,1)$ and is the expected returns.

## D.4 SUPPORT VECTOR MACHINE (SVM)

$$\min_x \quad x^\top x + \lambda \mathbf{1}^\top \boldsymbol{t}$$
$$\text{subject to} \quad \boldsymbol{t} \geq \mathbf{diag}(\boldsymbol{b})\boldsymbol{A}x + \mathbf{1}, \tag{A.15}$$
$$\boldsymbol{t} \geq 0,$$

**Problem instances** We set the matrix $\boldsymbol{A} \in \mathbb{R}^{m \times n}$, which has elements $\boldsymbol{A}_{ij}$ as follows:

$$A_{ij} = \begin{cases} \mathcal{N}(\frac{1}{n}, \frac{1}{n}), & i \leq \frac{m}{2} \\ \mathcal{N}(-\frac{1}{n}, \frac{1}{n}), & \text{otherwise} \end{cases}$$

We generate the random vector $\boldsymbol{b} \in \mathbb{R}^m$ ahead as follows:

$$b_i = \begin{cases} +1, & i \leq \frac{m}{2} \\ -1, & \text{otherwise} \end{cases}$$

we choose the scalar $\lambda$ as $\frac{1}{2}$.

## D.5 HUBER FITTING

$$\min_x \quad \boldsymbol{u}^\top \boldsymbol{u} + 2M\mathbf{1}^\top(\boldsymbol{r} + \boldsymbol{s})$$
$$\text{subject to} \quad \boldsymbol{A}x - \boldsymbol{b}\boldsymbol{u} = \boldsymbol{r} - \boldsymbol{s}, \tag{A.16}$$
$$\boldsymbol{r} \geq 0,$$
$$\boldsymbol{s} \geq 0,$$

**Problem instances** We set the matrix $\boldsymbol{A} \in \mathbb{R}^{m \times n}$ with element $\boldsymbol{A}_{ij} \sim \mathcal{N}(0,1)$ to have 50% nonzero elements. To generate the vector $\boldsymbol{b} \in \mathbb{R}^m$, we choose the random vector $\boldsymbol{v} \in \mathbb{R}^n$ ahead as follows:

$$v_i = \begin{cases} \mathcal{N}(0, \frac{1}{4}), & \text{with probability } p = 0.95 \\ \mathcal{U}[0, 10], & \text{otherwise} \end{cases}$$

Then, we set the vector $\boldsymbol{b} = A(v + \epsilon)$, where $\epsilon_i \sim \mathcal{N}(0, \frac{1}{n})$.

Table A.3: **Range of $n, m$ which uses for each problem class generation**

|  | Random QP | EqQP | Porfolio | SVM | Huber | Control | Lasso | En_Random QP |
|---|---|---|---|---|---|---|---|---|
| n | $10 \sim 15$ | $30 \sim 40$ | $50 \sim 60$ | $5 \sim 6$ | $5 \sim 6$ | $2 \sim 6$ | $5 \sim 6$ | $20 \sim 30$ |
| m | $10 \sim 15$ | $15 \sim 20$ | $5 \sim 6$ | $50 \sim 60$ | $50 \sim 60$ | $1 \sim 3$ | $50 \sim 60$ | $45 \sim 70$ |
| Relation $m$ & $n$ | $m = 3 \times n$ | $m = \lfloor \frac{n}{2} \rfloor$ | $m = 10 \times n$ | $m = \lfloor \frac{n}{10} \rfloor$ | $m = 10 \times n$ | $m = \lfloor \frac{n}{2} \rfloor$ | $m = 10 \times n$ | $m = \lfloor \frac{7n}{3} \rfloor$ |

## D.6 Optimal Control

$$\min_{x} \quad x_T^\top \boldsymbol{Q}_T x_T + \sum_{t=0}^{T-1} x_t^\top Q x_t + \boldsymbol{u}_t^\top \boldsymbol{R} \boldsymbol{u}_t$$
$$\text{subject to} \quad x_{t+1} = \boldsymbol{A} x_t + \boldsymbol{B} \boldsymbol{u}_t,$$
$$x_t \in \mathcal{X}, \boldsymbol{u}_t \in \mathcal{U},$$
$$x_0 = x_{\text{init}},$$

(A.17)

**Problem instances** We set the dynamic $\boldsymbol{A} \in \mathbb{R}^{n \times n}$ as $\boldsymbol{A} = \boldsymbol{I} + \Delta$, where $\Delta_{ij} \sim \mathcal{N}(0, 0.01)$. We generate the matrix $\boldsymbol{B} \in \mathbb{R}^{n \times m}$, whose element $\boldsymbol{B}_{ij} \sim \mathcal{N}(0, 1)$. We choose state cost $mQ$ as $\text{diag}(\boldsymbol{q})$, where $q_i \sim \mathcal{U}(0, 10)$ with 30% zeros elements in $vq$. We generate the input cost $\boldsymbol{R}$ as $0.1\boldsymbol{I}$. We set time $T$ as 5.

## D.7 Lasso

$$\min_{x} \quad \frac{1}{2} \boldsymbol{y}^\top \boldsymbol{y} + \gamma \boldsymbol{1}^\top \boldsymbol{t}$$
$$\text{subject to} \quad \boldsymbol{y} = \boldsymbol{A} x - \boldsymbol{b},$$
$$- \boldsymbol{t} \leq x \leq \boldsymbol{t},$$

(A.18)

**Problem instances** We set the matrix $\boldsymbol{A} \in \mathbb{R}^{m \times n}$, whose element $\boldsymbol{A}_{ij} \sim \mathcal{N}(0, 1)$ with 90% nonzero elements. To generate the vector $\boldsymbol{b} \in \mathbb{R}^m$, we set the sparse vector $\boldsymbol{v} \in \mathbb{R}^n$ ahead as follows:

$$v_i = \begin{cases} 0, & \text{with probability } p = 0.5 \\ \mathcal{N}(0, \frac{1}{n}), & \text{otherwise} \end{cases}$$

Then, we chose the vector $\boldsymbol{b} = Av + \epsilon$ where $\epsilon_i \sim \mathcal{N}(0, 1)$. We set the weighting parameter $\gamma$ as $\frac{1}{5} \|\boldsymbol{A}^\top b\|_\infty$.

For the seven problem types mentioned above, the range of variables $n$ and $m$ can be referred to in the Table A.3. The relationship between $n$ and $m$ is also described to facilitate understanding.

## D.8 Entire Random QP

$$\min_{x} \quad \frac{1}{2} x^\top \boldsymbol{P} x + \boldsymbol{q}^\top x$$
$$\text{subject to} \quad \boldsymbol{l} \leq \boldsymbol{A} x \leq \boldsymbol{u},$$
$$\boldsymbol{B} x = \boldsymbol{b},$$

(A.19)

**Problem instances** We set a random positive semidefinite matrix $\boldsymbol{P} \in \mathbb{R}^{n \times n}$ by using matrix multiplication on random matrix $\boldsymbol{M}$ and its transpose, whose element $\boldsymbol{M}_{ij} \sim \mathcal{N}(0, 1)$ has only 15% being nonzero elements. We generate the inequality constraint matrix $\boldsymbol{A} \in \mathbb{R}^{m_1 \times n}$ with $\boldsymbol{A}_{ij} \sim \mathcal{N}(0, 1)$ with only 60% nonzero elements. We chose upper bound $\boldsymbol{u} \in \mathbb{R}^{m_1}$ with $\boldsymbol{u}_i \sim (Ax)_i + \mathcal{N}(0, 1)$, where $x$ is a randomly chosen vector and $\boldsymbol{l}_i$ as $-\infty$. We generate the equality constraint matrix $\boldsymbol{B} \in \mathbb{R}^{m_2 \times n}$ with $\boldsymbol{B}_{ij} \sim \mathcal{N}(0, 1)$ with only 60% nonzero elements. We set constant $\boldsymbol{b} \in \mathbb{R}^{m_2}$, with $\boldsymbol{b}_i = (Bx)_i$, where $x$ is a randomly chosen vector at the generating process of vector $\boldsymbol{u}$. We set $m_1$, and $m_2$ as $\lfloor \frac{m}{7} \rfloor$ and $\lfloor \frac{6m}{7} \rfloor$, respectively.

Table A.4: **QP size generalization results in table format**. Smaller is better ($\downarrow$). Best in **bold**. We measure the average and standard deviation of the iterations for each QP sizes.

| | $10 \sim 15$ | $15 \sim 100$ | $100 \sim 200$ | $200 \sim 300$ | $300 \sim 400$ | $400 \sim 500$ | $500 \sim 600$ | $600 \sim 700$ | $700 \sim 800$ | $800 \sim 900$ | $900 \sim 1000$ |
|---|---|---|---|---|---|---|---|---|---|---|---|
| | | | | | Problem size ($N$) | | | | | | |
| CA-ADMM | **27.44** $\pm$ 4.86 | **31.70** $\pm$ 8.29 | **34.70** $\pm$ 3.07 | **40.80** $\pm$ 4.60 | **43.40** $\pm$ 2.24 | **46.70** $\pm$ 3.69 | **47.80** $\pm$ 2.56 | **49.10** $\pm$ 1.97 | **48.70** $\pm$ 2.00 | **50.10** $\pm$ 1.44 | **50.50** $\pm$ 1.28 |
| RLQP | 69.42 $\pm$ 27.35 | 101.44 $\pm$ 20.00 | 137.60 $\pm$ 20.97 | 169.40 $\pm$ 12.16 | 186.20 $\pm$ 8.42 | 214.60 $\pm$ 9.24 | 226.20 $\pm$ 5.19 | 234.20 $\pm$ 4.83 | 243.80 $\pm$ 7.08 | 252.40 $\pm$ 7.86 | 260.80 $\pm$ 16.15 |
| OSQP | 81.12 $\pm$ 21.93 | 102.82 $\pm$ 31.96 | 120.00 $\pm$ 22.43 | 143.00 $\pm$ 15.54 | 118.60 $\pm$ 15.54 | 137.80 $\pm$ 19.78 | 131.60 $\pm$ 13.18 | 142.20 $\pm$ 13.23 | 137.60 $\pm$ 15.11 | 130.80 $\pm$ 4.71 | 130.40 $\pm$ 9.73 |

## E    DETAIL OF THE EXPERIMENT RESULTS

### E.1    DETAILS OF THE IN-DOMAIN EXPERIMENT

In this section, we provide the problem generation, and additional experimental results for the In-domain experiments.

**Training data generation**    For the training set generation, the problems for each class are generated randomly with instance generating rules in (Appendix D). Each problem's size of QP is described in Table A.3.

**Evaluation data generation**    Table 1's evaluation data generated in the same way as training data generation. To construct empty intersection between train and test set, each set is generated from a different random seed. To verify the QP size generalization test (Fig. 3, Table A.4), problems are generated randomly for the problem class **Random QP** with dimension of $x$ in $[100, 1000]$.

**Additional results**    We provide the following Table A.4 to show the numerical results difficult to visualize in Fig. 3.

### E.2    DETAILS OF THE CROSS-DOMAIN EXPERIMENT

We provide the problem generation for the cross-domain experiments.

**Training data generation**    We create a new problem class named **Entire Random QP** that contains both the randomly generated inequality and equality constraints as there is no problem class in OSQP. The form of the QP problem class and instance generating rule are described in Appendix D.8. With the new problem class, we generate the training set with size in Table A.3.

**Evaluation data generation**    For the training set generation, the problems for each class are generated randomly with instance generating rules in (Appendix D). the size of each problem is described in Table A.3.

# F  ADDITIONAL EXPERIMENTS

## F.1  RESULT FOR UNIT COMPUTATIONAL TIME OF THE SIZE GENERALIZATION EXPERIMENT

Table A.5: **Unit computational time results for each QP sizes in table format**. Smaller is better (↓). Best in **bold**. We measure the average of the unit time per a MDP step for each QP sizes. The unit of every element is $Second$.

| Unit time per a MDP step | OSQP | | | | RLQP | | | | CA-ADMM | | | |
|---|---|---|---|---|---|---|---|---|---|---|---|---|
| Problem size | State Construction | linear system solving | $\rho$ computation | Total time | State Construction | linear system solving | $\rho$ computation | Total time | State Construction | linear system solving | $\rho$ computation | Total time |
| $15 \sim 100$ | 0.000 | 0.003 | 0.000 | 0.008 | 0.000 | 0.003 | 0.001 | 0.008 | 0.006 | 0.003 | 0.071 | 0.085 |
| $100 \sim 200$ | 0.000 | 0.008 | 0.000 | 0.014 | 0.000 | 0.008 | 0.001 | 0.015 | 0.008 | 0.010 | 0.101 | 0.127 |
| $200 \sim 300$ | 0.000 | 0.036 | 0.001 | 0.050 | 0.000 | 0.022 | 0.001 | 0.036 | 0.018 | 0.045 | 0.130 | 0.209 |
| $300 \sim 400$ | 0.000 | 0.079 | 0.001 | 0.102 | 0.000 | 0.046 | 0.001 | 0.070 | 0.026 | 0.093 | 0.154 | 0.300 |
| $400 \sim 500$ | 0.000 | 0.179 | 0.003 | 0.220 | 0.000 | 0.100 | 0.001 | 0.144 | 0.037 | 0.184 | 0.185 | 0.454 |
| $500 \sim 600$ | 0.000 | 0.324 | 0.005 | 0.392 | 0.000 | 0.163 | 0.001 | 0.229 | 0.055 | 0.340 | 0.218 | 0.690 |
| $600 \sim 700$ | 0.000 | 0.499 | 0.007 | 0.597 | 0.000 | 0.237 | 0.001 | 0.334 | 0.068 | 0.507 | 0.251 | 0.933 |
| $700 \sim 800$ | 0.000 | 0.746 | 0.010 | 0.881 | 0.000 | 0.374 | 0.001 | 0.510 | 0.087 | 0.743 | 0.265 | 1.250 |
| $800 \sim 900$ | 0.000 | 1.029 | 0.012 | 1.204 | 0.000 | 0.507 | 0.001 | 0.679 | 0.113 | 1.053 | 0.308 | 1.664 |
| $900 \sim 1000$ | 0.000 | 1.539 | 0.016 | 1.778 | 0.000 | 0.815 | 0.001 | 1.045 | 0.150 | 1.599 | 0.358 | 2.363 |

## F.2  RESULT FOR TOTAL COMPUTATIONAL TIME OF THE SIZE GENERALIZATION EXPERIMENT

Table A.6: **Total computational time results for each QP sizes in table format**. Smaller is better (↓). Best in **bold**. We measure the average of the total time and the MDP steps for each QP sizes.

| Problem size | OSQP | | | RLQP | | | CA-ADMM | | |
|---|---|---|---|---|---|---|---|---|---|
| | Total / MDP steps ($Second$) | MDP steps | Total time ($Second$) | Total / MDP steps ($Second$) | MDP steps | Total time ($Second$) | Total / MDP steps ($Second$) | MDP steps | Total time ($Second$) |
| $15 \sim 100$ | 0.008 | 11.2 | 0.087 | 0.008 | 10.4 | 0.082 | 0.085 | 3.6 | 0.306 |
| $100 \sim 200$ | 0.014 | 12.6 | 0.179 | 0.015 | 14.2 | 0.213 | 0.127 | 4.2 | 0.535 |
| $200 \sim 300$ | 0.050 | 14.6 | 0.734 | 0.036 | 17.6 | 0.636 | 0.209 | 4.8 | 1.001 |
| $300 \sim 400$ | 0.102 | 12.6 | 1.291 | 0.070 | 19.2 | 1.343 | 0.300 | 5.0 | 1.499 |
| $400 \sim 500$ | 0.220 | 14.4 | 3.172 | 0.144 | 22.0 | 3.158 | 0.454 | 5.0 | 2.269 |
| $500 \sim 600$ | 0.392 | 13.8 | 5.416 | 0.229 | 23.2 | 5.312 | 0.690 | 5.2 | 3.588 |
| $600 \sim 700$ | 0.597 | 14.6 | 8.715 | 0.334 | 24.0 | 8.012 | 0.933 | 5.6 | 5.226 |
| $700 \sim 800$ | 0.881 | 14.4 | 12.693 | 0.510 | 24.8 | 12.636 | 1.250 | 5.2 | 6.499 |
| $800 \sim 900$ | 1.204 | 13.8 | 16.609 | 0.679 | 25.8 | 17.527 | 1.664 | 5.8 | 9.654 |
| $900 \sim 1000$ | 1.778 | 13.2 | 23.472 | 1.045 | 26.8 | 28.007 | 2.363 | 5.6 | 13.233 |

## F.3  RESULT FOR LP EXPERIMENT

Table A.7: **LP experiment results.**: Smaller is better (↓). Best in **bold**. We measure the average and standard deviation of the iterations and average total computational time for Random LP problems.

| | CA-ADMM | RLQP | OSQP |
|---|---|---|---|
| Total time ($Second$) | 0.367 | 0.092 | 1.228 |
| Solve ratio[a] | 1.00 | 1.00 | 0.58 |
| Iterations | 79.94 | 139.28 | 2432.74 |

[a] Solve ratio is ratio between solved problems and total generated problems.

## F.4 RESULT FOR MAROS & MESZAROS PROBLEMS

Table A.8: **QP-benchamrk results.**(`Random QP + EqQP+ Portfolio+ SVM+ Huber+ Control+ Lasso`) → `X`, **where** `X` ∈ {`Maros & Meszaros`}: Smaller is better (↓). Best in **bold**. We measure the iterations and the Total Computation time for each QP benchmark.

| Instance | Problem information | | | Iterations | | | Computation time (sec.) | | |
|---|---|---|---|---|---|---|---|---|---|
| | $n$ | $m$ | nonzeros | CA-ADMM | RLQP | OSQP | CA-ADMM | RLQP | OSQP |
| AUG2D | 20200 | 30200 | 80000 | **16** | 41 | 28 | 0.617 | 0.878 | **0.528** |
| AUG2DC | 20200 | 30200 | 80400 | **16** | 33 | 27 | 0.551 | 0.668 | **0.46** |
| AUG2DCQP | 20200 | 30200 | 80400 | **82** | failed | 580 | **2.679** | failed | 11.133 |
| AUG2DQP | 20200 | 30200 | 80000 | **110** | failed | 574 | **3.518** | failed | 10.082 |
| AUG3D | 3873 | 4873 | 13092 | **19** | 42 | **19** | 0.32 | 0.179 | **0.072** |
| AUG3DC | 3873 | 4873 | 14292 | 17 | 33 | **15** | 0.24 | 0.145 | **0.062** |
| AUG3DCQP | 3873 | 4873 | 14292 | **28** | 38 | 37 | 0.39 | 0.152 | **0.129** |
| AUG3DQP | 3873 | 4873 | 13092 | **31** | 70 | 40 | 0.56 | 0.281 | **0.167** |
| CONT-050 | 2597 | 4998 | 17199 | **19** | failed | 32 | 0.273 | failed | **0.137** |
| CONT-100 | 10197 | 19998 | 69399 | 20 | **19** | 37 | 0.7 | **0.576** | 0.9 |
| CONT-101 | 10197 | 20295 | 62496 | **1141** | failed | 1607 | 53.217 | failed | **45.805** |
| CONT-200 | 40397 | 79998 | 278799 | **21** | 33 | 44 | **4.94** | 8.974 | 7.03 |
| CONT-201 | 40397 | 80595 | 249996 | 3761 | failed | **3154** | 817.087 | failed | **793.884** |
| CVXQP1_M | 1000 | 1500 | 9466 | **41** | 328 | 61 | 0.965 | 2.445 | **0.476** |
| CVXQP1_S | 100 | 150 | 920 | **28** | 56 | 50 | 0.335 | 0.042 | **0.035** |
| CVXQP2_L | 10000 | 12500 | 87467 | **27** | 212 | 43 | **11.636** | 78.098 | 18.781 |
| CVXQP2_M | 1000 | 1250 | 8717 | **25** | 175 | 41 | 0.378 | 0.53 | **0.13** |
| CVXQP2_S | 100 | 125 | 846 | **24** | 126 | 34 | 0.308 | 0.092 | **0.024** |
| CVXQP3_L | 10000 | 17500 | 102465 | **85** | failed | 95 | **306.795** | failed | 346.485 |
| CVXQP3_M | 1000 | 1750 | 10215 | 212 | failed | **153** | 5.46 | failed | **1.53** |
| CVXQP3_S | 100 | 175 | 994 | **24** | 51 | 38 | 0.368 | 0.047 | **0.031** |
| DPKLO1 | 133 | 210 | 1785 | **20** | 36 | 31 | 0.176 | 0.033 | **0.031** |
| DTOC3 | 14999 | 24997 | 64989 | failed | failed | **248** | failed | failed | **2.037** |
| DUAL1 | 85 | 86 | 7201 | **17** | 26 | 29 | 0.172 | 0.024 | **0.023** |
| DUAL2 | 96 | 97 | 9112 | **17** | 26 | 21 | 0.204 | 0.03 | **0.024** |
| DUAL3 | 111 | 112 | 12327 | **17** | 24 | 21 | 0.181 | 0.032 | **0.027** |
| DUAL4 | 75 | 76 | 5673 | **17** | 25 | 42 | 0.161 | **0.021** | 0.032 |
| DUALC1 | 9 | 224 | 2025 | **38** | 110 | 66 | 0.308 | 0.076 | **0.044** |
| DUALC2 | 7 | 236 | 1659 | **40** | 210 | 59 | 0.324 | 0.142 | **0.038** |
| DUALC5 | 8 | 286 | 2296 | **39** | 61 | 48 | 0.309 | 0.047 | **0.033** |
| DUALC8 | 8 | 511 | 4096 | **44** | 102 | 64 | 0.402 | 0.082 | **0.052** |
| EXDATA | 3000 | 6001 | 2260500 | **85** | 3004 | 86 | 53.737 | 1544.541 | **41.597** |
| GENHS28 | 10 | 18 | 62 | **13** | 27 | 14 | 0.151 | 0.019 | **0.01** |
| GOULDQP2 | 699 | 1048 | 2791 | **20** | 33 | **20** | 0.165 | 0.032 | **0.018** |
| GOULDQP3 | 699 | 1048 | 3838 | **20** | 23 | 34 | 0.17 | **0.026** | 0.037 |
| HS118 | 15 | 32 | 69 | 769 | 479 | **202** | 6.739 | 0.27 | **0.098** |

| name | Problem information | | | Iterations | | | Total Computation time (*Second*) | | |
|------|------|------|------|------|------|------|------|------|------|
| | n | m | nonzeros | CA-ADMM | RLQP | OSQP | CA-ADMM | RLQP | OSQP |
| HS21 | 2 | 3 | 6 | 22 | 45 | **21** | 0.193 | 0.031 | **0.016** |
| HS268 | 5 | 10 | 55 | **15** | 161 | **15** | 0.142 | 0.107 | **0.011** |
| HS35 | 3 | 4 | 13 | **17** | 19 | 29 | 0.143 | **0.013** | 0.019 |
| HS35MOD | 3 | 4 | 13 | 14 | 31 | **13** | 0.143 | 0.024 | **0.01** |
| HS51 | 5 | 8 | 21 | **18** | 32 | 19 | 0.167 | 0.025 | **0.023** |
| HS52 | 5 | 8 | 21 | 13 | 29 | **12** | 0.164 | 0.02 | **0.01** |
| HS53 | 5 | 8 | 21 | 17 | 31 | **13** | 0.146 | 0.02 | **0.009** |
| HS76 | 4 | 7 | 22 | **15** | 46 | 39 | 0.145 | 0.028 | **0.022** |
| HUES-MOD | 10000 | 10002 | 40000 | 1830 | 306 | **31** | 524.61 | **19.032** | 244.839 |
| HUESTIS | 10000 | 10002 | 40000 | 1830 | 306 | **31** | 522.189 | **18.395** | 246.329 |
| KSIP | 20 | 1021 | 19938 | 75 | 1993 | **53** | 1.385 | 6.166 | **0.053** |
| LASER | 1002 | 2002 | 9462 | **25** | 49 | 44 | 0.315 | 0.068 | **0.055** |
| LISWET1 | 10002 | 20002 | 50004 | **27** | 31 | 56 | 0.409 | **0.169** | 0.212 |
| LISWET10 | 10002 | 20002 | 50004 | **23** | 31 | 56 | 0.375 | **0.164** | 0.23 |
| LISWET11 | 10002 | 20002 | 50004 | **26** | 31 | 56 | 0.413 | **0.17** | 0.21 |
| LISWET12 | 10002 | 20002 | 50004 | **26** | 31 | 56 | 0.37 | **0.166** | 0.235 |
| LISWET2 | 10002 | 20002 | 50004 | **21** | 31 | 56 | 0.399 | **0.174** | 0.218 |
| LISWET3 | 10002 | 20002 | 50004 | **21** | 31 | 56 | 0.431 | **0.172** | 0.215 |
| LISWET4 | 10002 | 20002 | 50004 | **21** | 31 | 56 | 0.402 | **0.173** | 0.214 |
| LISWET5 | 10002 | 20002 | 50004 | 21 | **15** | 50 | 0.429 | **0.087** | 0.184 |
| LISWET6 | 10002 | 20002 | 50004 | **26** | 31 | 56 | 0.415 | **0.172** | 0.229 |
| LISWET7 | 10002 | 20002 | 50004 | **21** | 31 | 56 | 0.396 | **0.17** | 0.211 |
| LISWET8 | 10002 | 20002 | 50004 | **21** | 31 | 56 | 0.406 | **0.175** | 0.214 |
| LISWET9 | 10002 | 20002 | 50004 | **21** | 31 | 56 | 0.394 | **0.171** | 0.215 |
| LOTSCHD | 12 | 19 | 72 | **21** | 183 | 48 | 0.285 | 0.122 | **0.032** |
| MOSARQP1 | 2500 | 3200 | 8512 | **41** | 73 | 56 | 0.618 | 0.163 | **0.099** |
| MOSARQP2 | 900 | 1500 | 4820 | **24** | 131 | 55 | 0.331 | 0.204 | **0.069** |
| POWELL20 | 10000 | 20000 | 40000 | failed | 2438 | **1055** | failed | 8.85 | **3.873** |
| PRIMAL1 | 325 | 410 | 6464 | **23** | 59 | 31 | 0.913 | 0.083 | **0.047** |
| PRIMAL2 | 649 | 745 | 9339 | **14** | 72 | 30 | 0.614 | 0.184 | **0.062** |
| PRIMAL3 | 745 | 856 | 23036 | **15** | 242 | 26 | 0.794 | 1.662 | **0.182** |
| PRIMAL4 | 1489 | 1564 | 19008 | **17** | 815 | 24 | 0.769 | 9.351 | **0.312** |
| PRIMALC1 | 230 | 239 | 2529 | **58** | failed | failed | **0.501** | failed | failed |
| PRIMALC2 | 231 | 238 | 2078 | **160** | failed | failed | **1.335** | failed | failed |
| PRIMALC5 | 287 | 295 | 2869 | **225** | failed | 4764 | **2.153** | failed | 5.028 |
| PRIMALC8 | 520 | 528 | 5199 | **79** | failed | failed | **0.856** | failed | failed |
| Q25FV47 | 1571 | 2391 | 130523 | **202** | failed | 544 | 13.749 | failed | **8.678** |
| QADLITTL | 97 | 153 | 637 | 1611 | failed | **138** | 30.348 | failed | **0.079** |
| QAFIRO | 32 | 59 | 124 | **39** | 1104 | failed | **0.415** | 0.605 | failed |
| QBANDM | 472 | 777 | 3023 | **113** | failed | failed | **3.78** | failed | failed |
| QBEACONF | 262 | 435 | 3673 | failed | failed | **56** | failed | failed | **0.057** |
| QBORE3D | 315 | 548 | 1872 | failed | failed | **847** | failed | failed | **0.667** |
| QBRANDY | 249 | 469 | 2511 | **901** | failed | failed | **31.157** | failed | failed |
| QCAPRI | 353 | 624 | 3852 | **421** | failed | 829 | 13.648 | failed | **0.737** |
| QE226 | 282 | 505 | 4721 | **173** | 3100 | failed | 6.079 | **2.971** | failed |

| name | Problem information | | | Iterations | | | Total Computation time (*Second*) | | |
|---|---|---|---|---|---|---|---|---|---|
| | n | m | nonzeros | CA-ADMM | RLQP | OSQP | CA-ADMM | RLQP | OSQP |
| QETAMACR | 688 | 1088 | 11613 | failed | 3303 | **1290** | failed | 6.886 | **2.701** |
| QFFFFF80 | 854 | 1378 | 10635 | **131** | failed | 694 | 6.949 | failed | **3.034** |
| QFORPLAN | 421 | 582 | 6112 | failed | 1950 | **740** | failed | 2.654 | **1.021** |
| QGFRDXPN | 1092 | 1708 | 3739 | 1991 | failed | **33** | 40.741 | failed | **0.037** |
| QGROW15 | 645 | 945 | 7227 | failed | failed | failed | failed | failed | failed |
| QGROW22 | 946 | 1386 | 10837 | failed | failed | failed | failed | failed | failed |
| QGROW7 | 301 | 441 | 3597 | failed | failed | failed | failed | failed | failed |
| QISRAEL | 142 | 316 | 3765 | **49** | 2518 | 83 | 1.705 | 2.239 | **0.07** |
| QPCBLEND | 83 | 157 | 657 | 33 | **19** | 79 | 0.621 | **0.014** | 0.051 |
| QPCBOEI1 | 384 | 735 | 4253 | failed | failed | **87** | failed | failed | **0.11** |
| QPCBOEI2 | 143 | 309 | 1482 | failed | failed | **51** | failed | failed | **0.038** |
| QPCSTAIR | 467 | 823 | 4790 | failed | failed | **739** | failed | failed | **0.815** |
| QPILOTNO | 2172 | 3147 | 16105 | 361 | failed | **20** | 23.219 | failed | **0.152** |
| QPTEST | 2 | 4 | 10 | **12** | 43 | 33 | 0.138 | 0.031 | **0.024** |
| QRECIPE | 180 | 271 | 923 | **39** | 271 | 62 | 0.504 | 0.213 | **0.04** |
| QSC205 | 203 | 408 | 785 | 19 | **7** | 64 | 0.206 | **0.007** | 0.059 |
| QSCAGR25 | 500 | 971 | 2282 | failed | failed | **259** | failed | failed | **0.215** |
| QSCAGR7 | 140 | 269 | 602 | failed | failed | **543** | failed | failed | **0.331** |
| QSCFXM1 | 457 | 787 | 4456 | failed | failed | **382** | failed | failed | **0.387** |
| QSCFXM2 | 914 | 1574 | 8285 | failed | failed | **1176** | failed | failed | **1.729** |
| QSCFXM3 | 1371 | 2361 | 11501 | 1361 | failed | **729** | 57.445 | failed | **1.37** |
| QSCORPIO | 358 | 746 | 1842 | **69** | 348 | failed | 1.226 | **0.314** | failed |
| QSCRS8 | 1169 | 1659 | 4560 | **83** | failed | failed | **1.854** | failed | failed |
| QSCSD1 | 760 | 837 | 4584 | **41** | 118 | 215 | 0.799 | **0.195** | 0.331 |
| QSCSD6 | 1350 | 1497 | 8378 | **49** | 289 | 285 | 0.818 | **0.478** | 0.67 |
| QSCSD8 | 2750 | 3147 | 16214 | **54** | 2256 | 248 | 1.143 | 6.356 | **0.814** |
| QSCTAP1 | 480 | 780 | 2442 | **55** | 483 | failed | 0.91 | **0.464** | failed |
| QSCTAP2 | 1880 | 2970 | 10007 | **42** | 280 | failed | 0.987 | **0.757** | failed |
| QSCTAP3 | 2480 | 3960 | 13262 | **52** | 324 | failed | 1.468 | **1.063** | failed |
| QSEBA | 1028 | 1543 | 6576 | failed | failed | failed | failed | failed | failed |
| QSHARE1B | 225 | 342 | 1436 | failed | failed | **574** | failed | failed | **0.38** |
| QSHARE2B | 79 | 175 | 873 | **56** | failed | 2277 | **0.892** | failed | 1.276 |
| QSHELL | 1775 | 2311 | 74506 | failed | failed | **71** | failed | failed | **0.55** |
| QSHIP04L | 2118 | 2520 | 8548 | 123 | failed | **115** | 2.916 | failed | **0.363** |
| QSHIP04S | 1458 | 1860 | 5908 | **111** | failed | **111** | 2.713 | failed | **0.23** |
| QSHIP08L | 4283 | 5061 | 86075 | **125** | failed | 132 | 4.887 | failed | **1.375** |
| QSHIP08S | 2387 | 3165 | 32317 | 278 | failed | **178** | 9.257 | failed | **0.566** |
| QSHIP12L | 5427 | 6578 | 144030 | 741 | failed | **150** | 33.934 | failed | **1.872** |
| QSHIP12S | 2763 | 3914 | 44705 | failed | failed | **144** | failed | failed | **0.487** |
| QSIERRA | 2036 | 3263 | 9582 | failed | failed | **206** | failed | failed | **0.671** |
| QSTAIR | 467 | 823 | 6293 | **198** | failed | 353 | 6.397 | failed | **0.436** |
| QSTANDAT | 1075 | 1434 | 5576 | **88** | failed | failed | **2.129** | failed | failed |
| S268 | 5 | 10 | 55 | **15** | 161 | **15** | 0.138 | 0.1 | **0.01** |
| STADAT1 | 2001 | 6000 | 13998 | failed | failed | failed | failed | failed | failed |
| STADAT2 | 2001 | 6000 | 13998 | failed | failed | **558** | failed | failed | **0.979** |

| name | Problem information | | | Iterations | | | Total Computation time (*Second*) | | |
|---|---|---|---|---|---|---|---|---|---|
| | n | m | nonzeros | CA-ADMM | RLQP | OSQP | CA-ADMM | RLQP | OSQP |
| STADAT3 | 4001 | 12000 | 27998 | failed | failed | **702** | failed | failed | **2.145** |
| STCQP1 | 4097 | 6149 | 66544 | **22** | 110 | 39 | **6.154** | 19.142 | 6.738 |
| STCQP2 | 4097 | 6149 | 66544 | **21** | 109 | 40 | 1.529 | 3.519 | **1.191** |
| TAME | 2 | 3 | 8 | **19** | 26 | 21 | 0.138 | 0.019 | **0.016** |
| UBH1 | 18009 | 30009 | 72012 | 93 | 4757 | **92** | 3.348 | 39.363 | **0.787** |
| VALUES | 202 | 203 | 7846 | 34 | **18** | 37 | 0.682 | **0.019** | 0.031 |
| YAO | 2002 | 4002 | 10004 | 764 | **528** | 797 | 8.107 | **0.627** | 0.853 |
| ZECEVIC2 | 2 | 4 | 7 | **22** | 232 | 36 | 0.204 | 0.124 | **0.021** |

## F.5 MDP STEP INTERVAL ABLATION

Table A.9: **Step interval ablation results.**: Smaller is better (↓). Best in **bold**. We measure the average and standard deviation of the iterations for each step-intervals.

| Step-interval | Iterations | Computational time (sec.) |
|---|---|---|
| 5 | 27.02 | 0.194 |
| 10 | 30.42 | 0.111 |
| 50 | 53.39 | 0.065 |
| 100 | 65.72 | 0.050 |

