# OpenReview forum: "LEARNING CONTEXT-AWARE ADAPTIVE SOLVERS TO ACCELERATE QUADRATIC PROGRAMMING"
_ICLR.cc/2023/Conference — Submitted to ICLR 2023_

### Official Review · Reviewer_nZJ9 · 2022-10-22

**Confidence:** 5
**Clarity, Quality, Novelty And Reproducibility:** The paper is well-written and easy to…
**Correctness:** 3
**Technical Novelty And Significance:** 2
**Empirical Novelty And Significance:** 2
**Recommendation:** 5

**Strength And Weaknesses:**

Strengths:
1. This paper tests out a clear hypothesis that attention and temporal information can improve NN based parameter selection heuristics, and provide an empirical evaluation of their new approach.
2. The experiments are thorough and help ascertain the importance of each new contribution independently. The ablation study also does a reasonably good job of including necessary experiments for ascertaining the importance of different parameters and other aspects needed for reproducibility.

Weakness:
1. The technical contribution seems relatively incremental since both GRU and attention are commonly used. In addition, the intuition why GRU and attention can improve performance is not clear.
2. The authors only showed improvement in the iteration. How about the total time? In my opinion, using a graph neural network to get the parameter $\rho$ may increase the time complexity and resource consumption.
3. Why doesn't the author also apply similar algorithms for using ADMM to solve linear programming problems? If the idea works well for QP, it should work on LP as well. If not, providing reasons why it doesn't work on LP may help the community to understand the limitations and strengths of the proposed method.


**Summary Of The Paper:**

This paper introduces a machine learning based heuristic method to adaptively choose an important parameter of ADMM when applying to solve quadratic programming problems. The proposed method added a temporal component via a gated recurrent unit, which allows the model to incorporate consecutive information from previous iterations. In addition, the authors proposed heterogeneous graph attention for the embedding. The experiments show some improvements.

**Summary Of The Review:**

Overall, this paper has some non-trivial contributions but not significant ones. Thus it needs to be significantly improved to achieve the standard of ICLR.

=======After rebuttal=====

Thanks for answering my questions.

After reading your rebuttal, I still think using spatial and temporal with RL is not innovative. In addition,  only given the definition of contextual MDP is not enough to support the intuition of using the graph attention layer. Therefore, I still believe the contribution of this work is limited.

For the new experiments, it is only compared with QP ADMM methods by setting matrices to be zeros on some random data. I think it will be more convincing to compare with ADMM algorithms designed for LP on some real datasets.

Thus, I would like to maintain my original score.

---

### Official Review · Reviewer_L3nw · 2022-10-25

**Confidence:** 5
**Correctness:** 4
**Technical Novelty And Significance:** 3
**Empirical Novelty And Significance:** 3
**Recommendation:** 5

**Clarity, Quality, Novelty And Reproducibility:**

Clarity: the paper is well written and easy to read
Novelty: the proposed method is new with fair novelty


**Strength And Weaknesses:**

Strength:
i) The proposed method seems to have a substantial improvement over RLQP and OSQP in terms of number of iterations

Weakness:
i) The reported experimental results are on number of iterations. It would be much more meaningful to report the CPU time as well. In particular, the time for finding a good rho in CA-ADMM is supposed to take more time than that in OSQP. It is important to note if the decrease in number of iterations can compensate the overhead in finding a good rho.
ii) The paper is more on the empirical side. Therefore, it is very important to see the real performance of the proposed method. However, all the experiments are done on synthetic data sets (randomly generated). It is necessary to compare the proposed method with OSQP on public available real QP data set, for example, the Meszaros' QP set (http://old.sztaki.hu/~meszaros/public_ftp/qpdata/).
iii) If the paper only considers convex quadratic programming, please emphasize this in the title and abstract. There is no experiments reported on non-convex quadratic programming problems.


**Summary Of The Paper:**

This paper proposes a context-aware adaptive mechanism to adjust the step-size parameter rho in ADMM for solving convex quadratic programming problems, denote as CA-ADMM. It extracts the spatio-temporal context during the ADMM iterations. Numerical experiments on various type of QP sets are reported.

**Summary Of The Review:**

The paper proposes a new method for using ML techniques to accelerate ADMM for convex quadratic programming. The merit of this paper is more from the practical point of view. However, the reported experimental result has two main issues. First, the experiment is only conducted on synthetic/randomly generated data set. Second, only number of iterations is reported for comparing the efficiency between algorithms.

---

### Official Review · Reviewer_1ZMW · 2022-10-28

**Confidence:** 2
**Correctness:** 4
**Technical Novelty And Significance:** 3
**Empirical Novelty And Significance:** 3
**Recommendation:** 8

**Clarity, Quality, Novelty And Reproducibility:**

The paper is written well and easy to follow. The presented results seem novel and significant.

**Strength And Weaknesses:**

Strengths
- Novelty of the proposed architecture for ADMM updates
- The experimentatal results are strong.
-- The proposed method performs convincingly better than the baselines.
-- The dependency on the problem size is also studied in the experiments.
-- The ability to transfer trained model from smaller sizes to larger settings.


Questions
- (Minor) Table 2. What do the rows with labels "transferred" and "in-domain" refer? Some clarification may help understanding this better.
- Was there any experimentation done comparing how frequently \rho should be updated ? Or is there any objective justification for the update once in 10 iterations.
- How might the performances when we compare well-conditioned vs ill-conditioned matrix P?



**Summary Of The Paper:**

This paper considers the problem of QP in which the authors propose an adaptive scheme for ADMM step size selection (CA-ADMM). CA-ADMM obtains the step sizes using an MDP. The context for the MDP is arrived at using spatial and temporal means, wherein they use a graph neural network defined over the ADMM variables to generate the former and use the previous history to generate the latter. They do comprehensive experiments on various datasets illustrating that the proposed method leads to lesser number of admm iterations than the compared baselines.

**Summary Of The Review:**

The paper proposes a novel scheme for ADMM weights using an MDP based updates. The novelty of the scheme and the experimental results are convincing.

---

### Decision · Program_Chairs · 2023-01-20

**Decision:**

Reject

**Justification For Why Not Higher Score:**

Using spatial and temporal with RL is not innovative.
The intuition of using the graph attention layer is only motivated by giving the definition of the contextual MDP.
Concerning experiments, the proposed method is only compared with QP ADMM methods by setting matrices to be zeros on some random data. It could be more convincing to compare with ADMM algorithms designed for LP on some real datasets.

**Justification For Why Not Lower Score:**

It empirically show how attention and temporal information can improve NN based parameter selection heuristics.
The experiments are thorough, especially after wall clock time has been added.
The ablation study is of interest for ascertaining the influence of the parameters.


**Metareview: Summary, Strengths And Weaknesses:**

This paper proposes a machine learning inspired heuristic to adaptively choose an important parameter of ADMM when solvin quadratic convex programs. The proposed method added a temporal component via a gated recurrent unit, which allows the model to incorporate consecutive information from previous iterations.
In addition, the authors proposed heterogeneous graph attention for the embedding.
The experiments show some improvements in some situations.


**Summary Of Ac-Reviewer Meeting:**

No meeting: the reviewers never answered my emails.